# EPISODE TRANSFORMER: MODEL-BASED EPISODIC REINFORCEMENT LEARNING

## ABSTRACT

*Episodic Reinforcement Learning* (ERL) with movement primitives (MPs) has recently achieved significant success, especially in sparse and non-Markovian reward scenarios. By reasoning directly at the trajectory level via MPs, ERL results in smoother, energy-efficient policies and improved exploration capabilities for many real-world tasks. However, these black-box optimization approaches have very poor data-efficiency making them impractical for real-world applications. To mitigate this drawback, we propose *Episode Transformer*, a model-based ERL algorithm. Here, we learn a transformer-based episodic world model. To perform control we train a policy, with trust region constraints, purely in the world model's imagination. We compare our approach to state-of-the-art step-based and episodic RL methods on a variety of challenging robotic tasks under dense, sparse, and non-Markovian reward settings. The results show that the Episode Transformer is able to learn high-quality policies that retain all the benefits of previous deep ERL methods while requiring up to 5x fewer environment samples.

## 1 INTRODUCTION

Deep Reinforcement Learning (RL) has recently made some major breakthroughs in the field of artificial intelligence (Silver et al., 2016; Berner et al., 2019; Akkaya et al., 2019). Most recent advances in RL use step-based algorithms which means they repeatedly decide at each time step, which action to take given the state of the environment. When applied to robotic control, a step-based agent's policy is thus trained in raw action spaces like torques, joint angles, or end-effector positions, which forces the agent to make decisions at each time step of the trajectory. Consequently, step-based RL requires Markovian reward definitions. However, there are a lot of everyday tasks that require the agent to reason at a trajectory level rather than low-level raw step-based actions and that are difficult to describe as a dense, Markovian reward.

The field of Episodic reinforcement learning (ERL) tackles this problem of reasoning at an episode level by training policies that output parameterizations of an entire trajectory rather than each individual time step. Concretely, an ERL agent learns to select parameters for a high-level trajectory generator, such as a Movement Primitive Schaal et al. (2005); Paraschos et al. (2013), given a task description called the context. By directly reasoning at the trajectory level, ERL offers a powerful approach to addressing scenarios with sparse rewards, which are often easier and more intuitive to define. For example, if we want to achieve energy-efficient, time-optimal reaching motions, a sparse reward that only penalizes the distance to the goal in the last time step can be used. Since ERL allows for exploration directly in the trajectory space, and thus reasoning about the global episode-level task objective, the learning process is accelerated in such scenarios. In contrast, traditional step-based RL must explore a vast action space without receiving intermediate rewards and may struggle in sparse reward settings.

Similarly, ERL is more naturally suited for tasks with non-Markovian rewards. Consider for example the task of a robot throwing objects into bins placed at different locations. Here the agent needs to reason about the whole trajectory that it should take such that the object is imparted with the correct momentum to reach the bin. This task would require a non-Markovian reward definition because once the object is in the air, the robot can't influence its trajectory. Thus the reward received by the agent would depend on the entire trajectory and exploration at the trajectory level can result in smooth, energy-efficient policies. The same holds true for any task with inherent non-Markovian

optimality descriptors, resulting in non-Markovian rewards e.g., the maximum height during a jump or the minimum distance of a bat to a ball.

Recently, Deep Model-Free ERL (Bahl et al., 2020; Otto et al., 2023a) has shown superior performance over state-of-the-art step-based methods in solving sparse and non-Markovian reward tasks. However, data efficiency is a major downside for all of these methods. For instance, the approach described in Otto et al. (2023a) requires an excessive number of interactions with the robot, as many as $3 \cdot 10^8$ environment transitions for some tasks, which is often impracticable without informative prior knowledge.

Model-based reinforcement learning methods (Chua et al., 2018; Robine et al., 2023) have shown the potential to significantly reduce the sample complexity of model-free methods in step-based RL. In this work, we propose a model-based Episodic Reinforcement Learning algorithm, namely Episode Transformer, which learns a policy purely in the imagination of a transformer-based episodic world model. Our experiments on several challenging environments with different reward modalities (dense, sparse, and non-Markovian) show that the proposed approach can result in improved policies with up to 5x fewer environment interactions.

## 2 PRELIMINARIES

### 2.1 MOVEMENT PRIMITIVES

Movement Primitives (MPs) are commonly used for representing and learning basic movements in robotics, e.g., hitting and batting, grasping, etc. MP formulations allow for compact parameterizations of the robot's control policy, usually in the form of a weight vector $w$. Given the compact representation $w$, we use a trajectory generator $\psi(w)$ to transform from the parameter space to a trajectory $\tau$. Modulating the MP parameters permits imitation and reinforcement learning as well as adapting to different scenarios. There are several variants of MPs but they can be roughly split into two categories: Dynamic Movement Primitives (DMPs) (Schaal, 2006) and Probabilistic Movement Primitives (ProMPs) (Paraschos et al., 2013). The former uses a nonlinear dynamical system to represent the mean of a trajectory, while the latter represents not just a single trajectory but a probability distribution over trajectories by using a linear Gaussian basis function model. We use ProMP in this work to ensure a fairer comparison with ERL baselines, which use the same. However, DMP, ProMP, or more recent deep variants (Sekar et al., 2020; Li et al., 2023) of these can be used as abstract representations of trajectories in future extensions.

### 2.2 EPISODIC REINFORCEMENT LEARNING

In the contextual episodic reinforcement learning (ERL) framework, the agent is designed to solve a task at a trajectory level. Given a context $c$, the goal of ERL is to learn a policy that maximizes a reward function $R(\tau, c)$. However, as opposed to traditional step-based RL that outputs a low-level action, the ERL policy is optimized over a distribution of movement primitive (MP) parameters $w$. Since the MP parameter compactly represents an entire trajectory $\tau$, this provides a powerful mechanism to search directly over distributions of trajectories. The policy output $\pi_\theta(w|c)$, is thus a desired trajectory represented by the MP parameter, which is then executed in the actual environment by a tracking controller without any further agent input.

Here, the context vector $c$ characterizes the given task, for instance, given by a goal location or the location of an object, and the tracking controller can be implemented as a simple PD controller. The learning objective can thus be expressed as

$$\arg\max_\theta \mathbb{E}_{p(c)} \left[ \mathbb{E}_{\pi_\theta(w|c)}[R(w, c)] \right], \tag{1}$$

where $p(c)$ denotes the context distribution given by the task. Since there is no interaction or replanning by the agent during the episode, this setting is also referred to as black-box RL (BBRL). The return function $R(w, c)$ is not subject to any structural assumptions, and it can be any non-Markovian function of the resulting trajectory due to the black-box nature of the problem.

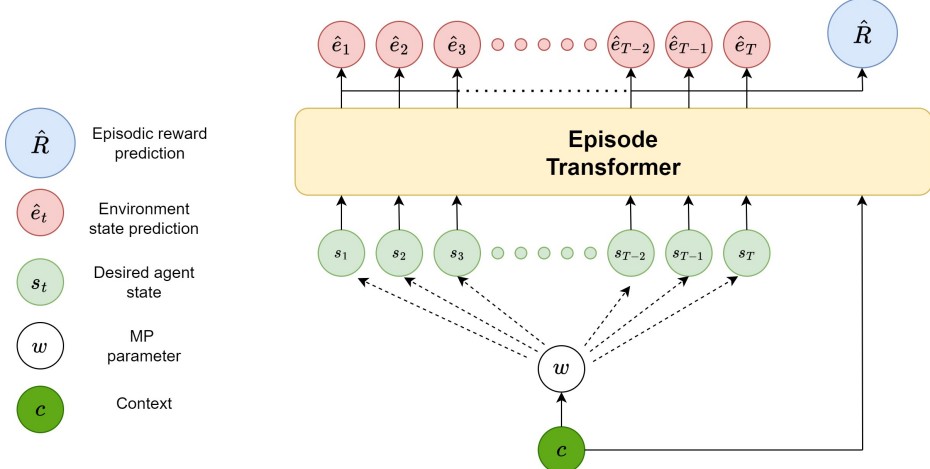

Figure 1: Schematic of Episode Transformer. We learn an episodic world model $g(c, \psi(w))$, which maps a context $c$ and abstract high-level action in the form of movement primitive weights $w$ to an imagination of the world states, $\hat{\tau} = \hat{e}_{1:T}$, along with an estimation of the cumulative returns ($\hat{R}$) over an entire episode. The context $c$ is usually a descriptor of the starting conditions in an episode along with the final goal.

## 3 THE EPISODE TRANSFORMER

Recent advances in model-free deep ERL algorithms (Bahl et al., 2020; Otto et al., 2023a), though powerful, come at the cost of sample inefficiency, making them impracticable for real-world robot learning. To mitigate this, we propose a model-based alternative, namely Episode Transformer.

**Model Based ERL Objective** As opposed to the model-free ERL objective in Equation 1, which assumes the cost/reward function $R(w, c)$ is a black box and unknown, we learn a model corresponding to the episodic reward $R_\phi(w, c)$, where $\phi$ corresponds to the parameters of the learned model. The learned model should approximate the true environment dynamics and the tracking controller behaviour to make reliable predictions of the episodic returns. We use a Transformer variant to learn this dynamics model and name the proposed method Episode Transformer. Given access to a parameterized model $R_\phi(w, c)$, the policy learning objective for model-based ERL is given as:

$$\arg \max_\theta \mathbb{E}_{p(c)} \mathbb{E}_{\pi_\theta(\boldsymbol{w}|c)} \left[ R_\phi(c, \boldsymbol{w}) \right]. \tag{2}$$

**Agent components** Similar to Hafner et al. (2019a), our agent learns behaviours purely in the imagination of the learnt model and has three components, (i) model learning, (ii) behaviour learning, and (iii) environment interaction. As detailed in Algorithm 1, Episode Transformer performs the following operations throughout the agent's lifetime, either interleaved or in parallel:

- Learning a model to predict the episodic cost/reward function $R_\phi(w, c)$, which implicitly requires knowledge of the forward dynamics of agent-environment interaction and low-level tracking controllers. We use a transformer-based non-autoregressive architecture to achieve this. The model objective and architecture are detailed in Section 2.2.

- Learning to reason with abstract high-level actions at a trajectory level using movement primitive based policies $\pi_\theta(w|c)$, as described in Section 3.2. The episodic policy is updated by propagating gradients of episodic cost/reward estimates back through the transformer dynamics.

- Executing the learned policy in the environment to collect new experiences and expand the model dataset.

## 3.1 EPISODIC WORLD MODELS WITH TRANSFORMERS

**Model Architecture** We use a Transformer (Vaswani et al., 2017) to build the episodic world model, $g_\phi(c, \psi(w))$, and name it Episode Transformer. As opposed to traditional step-based world models (Deisenroth and Rasmussen, 2011; Hafner et al., 2019a), which work with low-level actions, the episodic world model makes predictions about the reward/states of the world as a function of abstract representations $w$ of the desired episodic trajectory $\tau$ and context $c$. Concretely, the transformer model is tasked to predict a scalar corresponding to the cumulated episodic return $\hat{R} = R_\phi(w, c)$ and a vector of the actual environment states $\hat{\tau} = \hat{e}_{1:M}$, given the context $c$ and agent's desired trajectory, $\tau$, where $\tau = \psi(w) = s_{1:N}$. Thus we learn a model,

$$\hat{R}, \hat{\tau} = g_\phi(c, \tau). \tag{3}$$

Here $\hat{e}_t$ corresponds to the prediction of the actual state of the environment at time t, which may include actual states of both the agent and objects in the environment. $s_t$ corresponds to the desired trajectory of the agent at time t and comprises the joint positions ($q_t$) and velocities ($\dot{q}_t$) at time t.

The mapping in Equation 3 is, in essence, a sequence-to-sequence learning problem similar to the machine translation domain for which the original autoregressive version of the Transformer was proposed. Thus, it would only be natural to adopt their encoder-decoder (ED) architecture to our problem. However, since we evaluate the model for every policy update, fast inference is crucial to the wall-clock speed of the algorithm. The autoregressive prediction of the ED architecture needs one forward pass of the encoder but $T$ forward passes of the decoder. Since the dot-product attention operation itself already scales quadratically in $T$, the overall inference has a complexity of $\mathcal{O}\left(T^3\right)$. This massively slows down inference speed. Furthermore, when using the reparameterization trick as explained in 3.2, the analytical gradients from the reward prediction need to be propagated through the entire autoregressive inference in ED architecture. This essentially leads to a backpropagation-through-time (BPTT) scheme that further slows down the training and suffers from problems like vanishing gradients as well as local optima (Pascanu et al., 2013).

For these reasons, we instead implement our model as an encoder-only **non-autoregressive** architecture, i.e. we predict the entire output sequence at a single shot and do not employ any causal masking. This leads to significantly faster training and inference as we only need one forward and one backward pass during the policy update. The architecture is shown in Figure 1.

**Input Transformation** We project states at each time step $s_t$ from the desired agent trajectory, $\tau$, to an embedding dimension $d_{\text{model}}$ using a learned linear projection. We use a learned embedding matrix (Radford et al., 2018) as positional encodings of the time steps and add them to the state embeddings. Finally, we normalize the inputs using layer normalization and prepend the likewise projected context c. The transformed inputs are passed to the Transformer encoder.

**Prediction Head and Loss Functions** The model uses two different heads to predict its outputs. The environment state prediction head is implemented as a two-layer MLP that predicts the trajectory of actual environment states, $\hat{\tau} = \hat{e}_{1:M}$. We use a smooth L1-loss between the prediction and ground truth for this head, as shown below

$$l_1^{\text{smooth}}(x, y) = \begin{cases} 0.5(x-y)^2, & \text{if } |x-y| < 1, \\ |x-y| - 0.5, & \text{else} \end{cases}$$

The smooth L1-loss combines a quadratic loss (if the prediction is close to the target) with a linear one (if they are further apart. It is less sensitive to outliers compared to MSE Loss and results in stabler training. For the reward prediction head, we predict the entire episode return at once using and MLP. The network consists of one hidden layer of size $d_{\text{model}}$ and two layers that aggregate the transformer outputs of shape $(T, d_{\text{model}})$ to a scalar, first over the model dimension, then over the sequence length. We use $l_1^{\text{smooth}}$ as a loss function here as well. The model learning loss for a single episode/trajectory is thus formulated as follows:

$$L_{model} = l_1^{\text{smooth}}(\hat{R}, R) + \sum_{t=1}^{T} l_1^{\text{smooth}}(\hat{e}_t, e_t). \tag{4}$$

Here, R is the ground truth episodic returns, while $e_t$ is the ground truth environment states at time t. The extension to multiple trajectories is straightforward and omitted to keep the notations uncluttered.

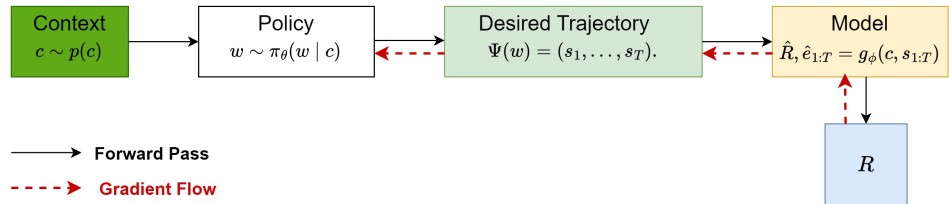

Figure 2: The policy update in the proposed model-based ERL framework. Episode Transformer learns an MP-based episodic policy by propagating analytic gradients of the predicted negative returns back through the learned transformer-based episodic dynamics model.

## 3.2 POLICY LEARNING

Given that we have access to the model of episodic returns $R_\phi(w, c)$, we learn a contextual controller that maximizes the expected return based on objective 2. To solve this objective, model-free deep ERL Otto et al. (2023a); Bahl et al. (2020) methods rely on policy gradient methods, that use the log-ratio trick to estimate the gradients of the objective. However, since in our case the model $R_\phi(w, c)$ is known and differentiable, the gradients can be estimated via the reparameterization trick (Kingma and Welling, 2013). This enables us to learn the policy in an end-to-end manner similar to Hafner et al. (2019b), where the analytic gradients of the estimated returns are propagated through the transformer dynamics and back into the policy. The end-to-end learning also results in a stronger learning signal with less variance as opposed to high-variance policy gradient objectives based on log-ratio tricks, which we verify empirically in an ablation in Section 5.3.

Concretely, during the policy learning stage, we sample a batch of contexts as inputs to the policy which outputs a Gaussian distribution over MP-parameters $\pi_\theta(w|c)$. We sample from this distribution using the reparameterization trick and unroll the sampled MP parameter $w$ to a desired trajectory using trajectory generator $\tau^d = \psi(w)$. Theoretically, it would also be possible to sample multiple trajectories from each policy distribution in the batch, effectively multiplying the batch size. However, we found this extension to not have much effect in practice while significantly reducing the training speed due to the additional computation. We then estimate the episodic return using the model $R_\phi(w, c)$, given the context and desired trajectory ($\tau_d$) as input. We use the negative return as a loss, which is back-propagated through the transformer model, back into the policy in an end-to-end manner. Policy learning thus does not involve any direct environment interactions and purely happens in the imagination of the learned model.

**Trust Region Constraints** Training this type of episodic policy in compact parameter ($w$) spaces poses unique challenges not present in the step-based RL methods. In the step-based case, smaller errors during action selection can still be corrected at a later time step. In ERL, however, since we only select one parameter vector $w$ per episode (and execute the corresponding trajectory with a low-level controller), no error correction of these parameters is possible. This necessitates policies with a higher level of precision. Thus, we build on the insights and method of Otto et al. (2023a) and enforce context-wise trust-region constraints via differentiable trust region layers (Otto et al., 2021). Since we learn from off-policy data, we do not have a sampling policy which we could as the natural choice for the constraining distribution $\pi_{old}$. Instead, we adopt an idea from Nachum et al. (2017) and keep a copy of the policy network whose parameters $\theta_{old}$ are a lagged geometric mean of the actual policy parameters $\theta$ that are updated after each training step

$$\theta_{old} \leftarrow \alpha\theta_{old} + (1 - \alpha)\theta, \tag{5}$$

where $\alpha$ is the rate of exponential decay. Thus, the TRPL penalizes divergence from a policy that is on average $1/(1 - \alpha)$ training steps in the past. The full objective of our algorithm is given as

$$\arg\min_\theta \mathbb{E}_{p(c)}\Big[\mathbb{E}_{\tilde{\pi}(w|c,\theta)}\big[-R_\phi(c, w)\big] + \beta \mathrm{KL}(\tilde{\pi}(\cdot|c, \theta)||\pi_\theta(\cdot|c)\Big], \tag{6}$$

where $\tilde{\pi}(w|c, \theta)$ is the TRPL-projection of the policy distribution $\pi_\theta(w|c)$ and $\beta$ is the weight of the trust region loss. We empirically find this extension to be crucial for successful learning and present an ablation in Figure 5.

## 4    RELATED WORKS

### 4.1    EPISODIC REINFORCEMENT LEARNING

Episodic RL has been popular in the robot learning community since it enables the learning of smooth, safe, reusable and energy-efficient skills. These properties are critical in real-world robot learning and control. Furthermore, they offer superior performance in several commonly used tasks in robotics that involve sparse and non-Markovian rewards compared to step-based counterparts (Schulman et al., 2015; 2017; Haarnoja et al., 2018; Otto et al., 2021). Most existing works in ERL (Schaal et al., 2005; Schaal, 2006; Peters et al., 2010; Abdolmaleki et al., 2015; Bahl et al., 2020; Otto et al., 2023a) focus on model-free versions where the forward dynamics and reward/cost function are assumed to be unknown and suffer from the major drawback of sample inefficiency and hence making them impractical in real robot learning. Kupcsik et al. (2013) proposed using Gaussian Process-based episodic forward models to learn model-based alternatives to ERL. However, unlike Kupcsik et al. (2013), we use much more expressive and generic Transformer world models and update our neural network policy using stochastic backpropagation instead of stochastic search. Moreover, these approaches only consider a linear mapping from context to parameter space while our method models a highly nonlinear context-parameter relationship using deep learning.

### 4.2    MODEL-BASED REINFORCEMENT LEARNING

The idea of training policies in a learned world modelwas first investigated in tabular environments (Sutton and Barto, 2018). There have been several advances since then that learn policies in the imagination of deep learning-based dynamics models (Ha and Schmidhuber, 2018; Hafner et al., 2019a) to learn sample efficient policies compared to their model-free counterparts. However, most of these models work at the level of fine-grained low-level actions, both in terms of dynamics and policy learning. Thus, they suffer from the same drawbacks as model-free step-based RL. In contrast, we learn an episodic world model with a powerful Transformer architecture that allows reasoning at a trajectory level using movement primitives. Thus, we combine the sample efficiency of model-based methods with improved exploration and reasoning in parameter space.

### 4.3    REINFORCEMENT LEARNING WITH TRANSFORMERS

There has been exciting progress recently in treating RL as a sequence modelling problem. Trajectory Transformer (Janner et al., 2021) use GPT-like autoregressive transformers to continuous control problems by discretizing state and action spaces and predicting the next states, actions and rewards in a sequence. They do not use any planning methods but utilize the beam search algorithm (Freitag and Al-Onaizan, 2017) from NLP domain instead. Decision Transformer (Chen et al., 2021) uses the same idea but only predicts the next actions and does not rely on discretization. Both these methods exhibit state-of-the-art performance on several standard offline RL benchmarks. Zheng et al. (2022) extended Decision Transformers to online settings. However, our method, Episode Transformer poses unique advantages that render them much more suitable for real-time robotic control. (i) As opposed to Janner et al. (2021); Chen et al. (2021) we do not rely on Transformer models during inference, thus allowing for computationally efficient and lightweight deployment during real-time inference. (ii) As opposed to working with low-level actions we reason with movement primitives giving them unique advantages of model-based Episodic RL like inducing smooth and energy-efficient control which is critical for robotics.

## 5    EXPERIMENTS

For our evaluation, we begin by demonstrating the effectiveness of our method in handling sparse rewards with significantly fewer environmental interactions. Afterwards, we show its ability to solve more challenging control problems that are difficult to solve in the step-based setting. Lastly, we validate the design decisions of our algorithm through ablations. We compare Episode Transformer to three main categories of algorithms: (i) **Episodic RL methods** like BB-TRPL, BB-PPO (Otto et al., 2023a) as well as the linear adaption method CMORE (Tangkaratt et al., 2017). (ii) **Step based RL methods** including PPO (Schulman et al., 2017), TRPL (Otto et al., 2021) and SAC (Haarnoja et al., 2018). (iii) **Methods that combine episodic and step-based RL** like NDP (Bahl et al., 2020)

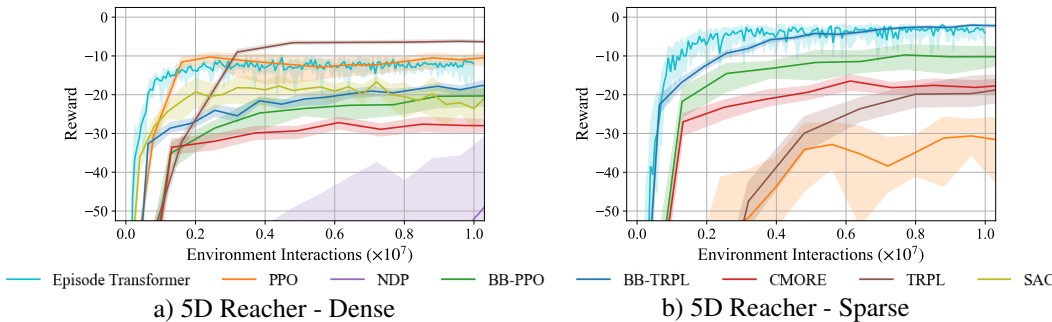

Figure 3: Learning curves for the 5D reacher with dense (left) and sparse (right) rewards. Episode Transformer is more sample efficient than BB-TRPL and BB-PPO in both reward settings while also achieving a better final performance on dense rewards.

from recent literature, which embed the structure of dynamical movement primitives (DMPs) into deep policies by reparametrizing action spaces via second-order differential equations. For all pure Episodic RL baselines, we use ProMPs as movement primitives to ensure a fair comparison.

We report results based on the total number of environment interactions for fair comparisons of their sample efficiency. Episodic approaches receive the context vector $c$ as a descriptor of the starting conditions and the goal of the episode. More concretely, the context consists of a subset of the observation space that is randomly initialized at the beginning of the episode, such as object and goal positions. The performance metric used is the undiscounted return over a full trajectory. We assessed our approach across 12 distinct random seeds, periodically calculating metrics by averaging the results over 10 evaluation episodes. We follow Agarwal et al. (2021) in reporting the *interquartile mean* (IQM) with a 95% stratified bootstrap confidence interval. For detailed descriptions of the environments and hyperparameters used, please refer to Appendix B and C, respectively.

### 5.1 REINFORCEMENT LEARNING UNDER DENSE AND SPARSE REWARDS

We start by benchmarking our approach to a simpler task on a modified version of the reacher agent from OpenAI Gym (Brockman et al., 2016). The agent uses 5 actuator joints but a limited context space, i.e. the location of the goal is restricted to $y \geq 0$. We investigate two types of rewards: a dense reward equivalent to the original reacher and a sparse reward that only provides the distance to the goal in the last episode time step. Although the reacher task has been solved by traditional step-based RL methods with a dense reward, the sparse reward setting has some advantages. (Otto et al., 2023a) argues for the use of sparse rewards, since they induce policies that are slower, precise and energy-efficient as opposed to dense rewards policies that encourage high-energy motions, and sacrifice accuracy for speed.

Figure 3 shows that Episode Transformer is more sample efficient than their model-free ERL counterparts like BB-TRPL and BB-PPO in both dense and sparse reward settings. Our method also achieves a better final performance on dense rewards. Step-based baselines like PPO and TRPL achieve a slightly better asymptotic performance than ours in the dense setting but are unable to consistently reach the goal with sparse reward signal. SAC solves the task partially in the dense setting but fails in the sparse setting. CMORE performs reasonably well, however, is able to cover only part of the context space due to its linear adaption strategy. Notably, NDP fails in both dense and sparse reward settings.

Now we demonstrate the utility of the proposed method in a more complex setting of a box-pushing task with a 7 DoFs robot arm. Here, the robot is tasked to precisely push and rotate a box to a goal position and orientation. As shown in Figure 4, under dense rewards, all methods except for SAC and NDP manage to solve the task eventually but Episode Transformer completes the task with fewer environment interactions and achieves a higher success rate. When it comes to the sparse reward setting, conventional step-based methods like PPO, TRPL and SAC show a significant degradation in performance as expected, while model-free ERL methods maintain a reasonable performance. Episode Transformer clearly outperforms all baselines both in terms of sample efficiency and success

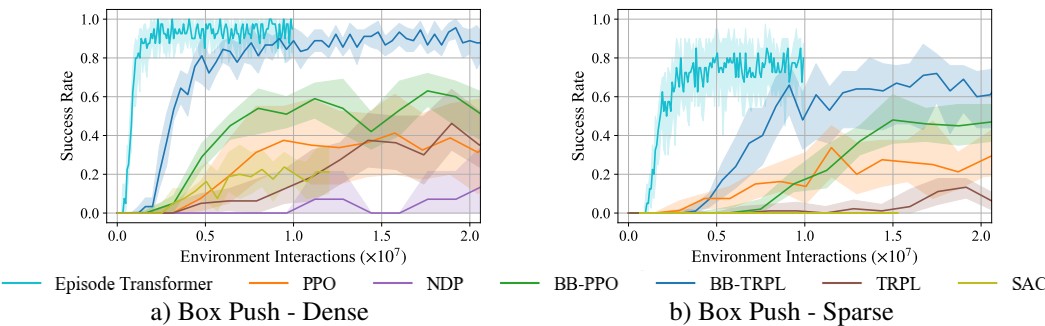

a) Box Push - Dense             b) Box Push - Sparse

Figure 4: Under dense rewards (left), all methods except for SAC and NDP manage to solve the task at least partially but Episode Transformer needs fewer interactions to do so. Step-based methods struggle under sparse rewards (right). Episode Transformer achieves the highest success rate while needing the fewest interactions.

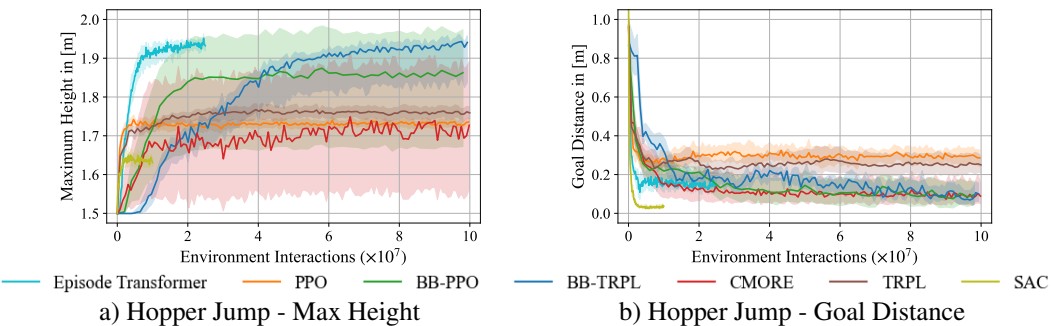

a) Hopper Jump - Max Height           b) Hopper Jump - Goal Distance

Figure 5: The maximum jumping height of the hopper's center of mass (left) and the target distance (right). ERL methods using the non-Markovian reward are able to jump up to 20cm higher than step-based methods. Episode Transformer only needs a fifth of the environment steps that BB-TRPL needs to reach the same maximum height.

rate. By learning a model of the environment and training the policy in its imagination, we can significantly reduce the number of environmental interactions required to solve the task making it a viable option for real-world robotics tasks.

## 5.2 DEALING WITH NON-MARKOVIAN REWARDS

Non-Markovian rewards are particularly useful for complex robot learning tasks, where the whole trajectory history is needed to provide feedback to the agent. We use a modification of the hopper from OpenAI Gym (Brockman et al., 2016), where we aim to jump as high as possible and land at a target location. The non-Markovian reward is defined as the maximum height and the minimum distance to the target achieved during the episode. Since step-based RL methods cannot handle such a reward formulation, we train them with a Markovian version of the reward that provides height and goal distance in each time step. While most methods manage to jump close to the target, only the ERL methods trained with the non-Markovian reward reach a good jumping height (Figure 5). Since the Markovian reward incentivizes height at each time step, step-based methods try to maximize it with multiple small jumps. In contrast, reasoning over the entire trajectory allows ERL methods to charge up for one big jump that achieves a higher maximum height. Among ERL methods, Episode Transformer and BB-TRPL reach the highest jumping height. However, our method achieves similar quality results with 5x fewer environment interactions.

## 5.3 ABLATIONS

Our ablation study evaluates our the design decisions and aims to answer the following questions:

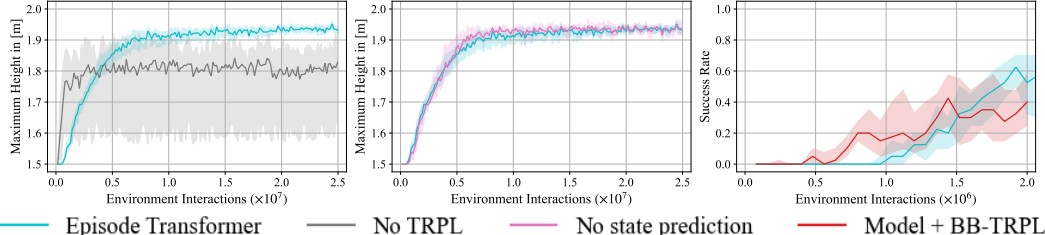

Figure 6: Ablations for the Episode Transformer: Without carefully constraining the policy learning with a trust region, the Episode Transformer converges to sub-optimal policies (left). Prediction of the trajectory of actual states does not improve performance in comparison to purely predicting the episode return (middle). Using the same model but training the policy with BB-TRPL with REINFORCE gradients lowers the success rate on box pushing.

**1. Can we learn policies without Trust Region Constraints?** We train the Episode Transformer without the trust region constraint imposed by the TRPL (Otto et al., 2021). In order to encourage sufficient exploration, we instead add an entropy regularization term to the policy objective. The results shown in Figure 6 (left) indicate that trust region constraints are crucial for stable learning. Without trust regions, the agent initially learns to achieve a decent height more quickly but converges to a sub-optimal solution. Additionally, the wider confidence interval indicates unstable learning and a higher variance across random seeds.

**2. What is the impact of the auxiliary state trajectory prediction objective?** We compare the effect of predicting the trajectory of actual environment states $e_{1:T}$ as an auxiliary objective to a version that only predicts the episode return. As Figure 6 (middle) demonstrates, the impact of the additional state prediction on the agent's performance is minimal. This result suggests that the information contained in the return signal and the desired trajectory $\tau^d$ is enough to learn a good mapping from the desired trajectory to episode return.

**3. Is there a benefit in using stochastic backpropagation over REINFORCE-style gradients?** We learn the same Transformer-based episodic world model described in Section 3.1 but update the policy on imagined trajectories using the on-policy BB-TRPL algorithm Otto et al. (2023a) that uses REINFORCE style gradient computation. As seen in Figure 6 (right) our end-to-end learnt method using stochastic backpropagation results gives better performance as opposed to high-variance policy gradient methods based on log-ratio tricks.

## 6  CONCLUSION AND LIMITATIONS

In this work, we proposed a model-based Deep ERL algorithm named Episode Transformer. We learn episodic policies with MPs purely in the imagination of a Transformer-based episodic world model. This allows us to keep all the benefits of deep ERL approaches yet significantly improve the data-efficiency making them a promising candidate for real-world robot learning tasks. Our experiments on a variety of benchmarks validate our hypothesis that Episode Transformers can learn high-quality policies both under dense and sparse reward settings. The utility of our approach is more pronounced under sparse and non-Markovian rewards where we achieve similar or higher quality solutions with 5x fewer environment interactions compared to model-free ERL methods.

The main limitation of our method is that the desired trajectory is planned in advance at the beginning of the episode, and hence, cannot be altered during the execution. This might be problematic for unforeseen events or perturbations, i.e. highly complex or reactive behaviour that cannot directly be modelled with the current motion representation. Recent work by Otto et al. (2023b) addresses this limitation by using a more sophisticated MP (ProDMPs, Li et al. (2023)) that allows for changing parameters during execution to incorporate replanning into ERL. We believe that integrating this to our approach would be a natural extension of our work. For future work, we will also investigate sequencing multiple MPs to solve complex long-horizon tasks involving sub-goals.

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

## A    PSEUDOCODE

```
def train():
    𝒟_model = collect_episodes_randomly(n_seed)
    for n_pretrain:
        update_model(𝒟_model)
    for n_train − n_seed:
        𝒟_model += collect_episodes(n_collect)
        for n_updates:
            episode_batch = 𝒟_model.sample_batch_uniformly()
            update_model(episode_batch)
            update_policy(episode_batch.context)

def update_policy(c):
    action_dist = policy(c)
    old_action_dist = old_policy(c)
    action_dist, L_KL = kl_projection(action_dist, old_action_dist)
    w = action_dist.rsample()
    τ = get_trajectory(w)
    R = model(c,τ)
    loss = −R + βL_KL
    loss.mean().backward()
    old_policy.update_params(policy.params())

def update_model(episode_batch):
    c, s_{1:T}, e_{1:T}, R = episode_batch
    context_emb = embed_context(c)
    state_emb = embed_state(s_{1:t})
    time_enc = embed_time(range(0, T))
    input_embeds = concat(context_emb, state_emb + time_enc)
    hidden_states = transformer_encoder(input_embeds)
    ê_{1:T} = states_head(hidden_states)
    R̂ = reward_head(hidden_states)
    loss = L_states(ê_{1:T}, e_{1:T}) + L_R(R̂, R)
    loss.mean().backward()
```

Listing 1: Training procedure of the Episode Transformer in pseudocode.

## B    ENVIRONMENTS

In order to demonstrate the effectiveness of our algorithm, we designed a series of experiments in three challenging robot control environments using the physics simulation MuJoCo (Todorov et al., 2012). The environments are the same ones as used in Otto et al. (2023b) but we include them here for completeness' sake.

### B.1    5D-REACHER

**Task Description.**  For the Reacher task, we modified the original OpenAI gym Reacher-v2 by adding three additional joints, resulting in a total of five joints. The task goal is still to minimize the distance between the goal point $p_{\text{goal}}$ and the end-effector $p$. We, however, only sample the goal point for $y \geq 0$, i. e. in the first two quadrants, to slightly reduce task complexity while maintaining the increased control complexity. The setup is shown in Figure 7 (top). The observation space remains unchanged, unless for the sparse reward where we additionally add the current step value to make learning possible for step-based methods. The context space only contains the coordinates of the goal position. The action space is the 5d equivalent to the original version. For the reward the original setting leverages the goal distance

$$R_{\text{goal}} = ||p - p_{\text{goal}}||_2, \tag{7}$$

and the action cost

$$\tau_t = \sum_i^K (a_t^i)^2. \tag{8}$$

**Dense Reward.** The dense reward in the 5d setting stays the same and the agent receives in each time step $t$

$$R_{\text{tot}} = -\tau_t - R_{\text{goal}}. \tag{9}$$

**Sparse Reward.** The sparse reward only returns the task reward in the last time step $T$ and additionally adds a velocity penalty $R_{\text{vel}} = \sum_i^K (\dot{q}_T^i)^2$, where $\dot{q}$ are the joint velocities, to avoid overshooting

$$R_{\text{tot}} = \begin{cases} -\tau_t & t < T, \\ -\tau_t - 200 R_{\text{goal}} - 10 R_{\text{vel}} & t = T. \end{cases} \tag{10}$$

### B.2 BOX-PUSHING

**Task Description.** The goal of the box-pushing task is to move a box to a specified goal location and orientation using the seven DoF Franka Emika Panda robotic arm. The basic setup is shown in Figure 7 (middle). The context space for this task is the goal position $x \in [0.3, 0.6]$, $y \in [-0.45, 0.45]$ and the goal orientation $\theta \in [0, 2\pi]$. In addition to the contexts, the observation space contains information about joints and the end-effector as well as the current box location and orientation. In each time step, we add gravity compensation to the original torque from the policy. The task is considered successfully solved if the position distance is $\leq 0.05$m and the orientation error is $\leq 0.5$rad.

**Reward Formulation.** For the total reward we consider different sub-rewards. First, the distance to the goal

$$R_{\text{goal}} = ||p - p_{\text{goal}}||, \tag{11}$$

where $p$ is the box position and $p_{\text{goal}}$ the goal position. Second, the rotation distance

$$R_{\text{rotation}} = \frac{1}{\pi} \arccos |r \cdot r_{\text{goal}}|, \tag{12}$$

where $r$ and $r_{\text{goal}}$ are the box orientation and goal orientation as quaternions, respectively. Third, an incentive to keep the rod within the box

$$R_{\text{rod}} = \text{clip}(||p - h_{\text{pos}}||, 0.05, 10), \tag{13}$$

where $h_{\text{pos}}$ is the position of the rod tip. Fourth, a similar incentive that encourages maintaining the rod in a desired rotation

$$R_{\text{rod\_rotation}} = \text{clip}(\frac{2}{\pi} \arccos |h_{\text{rot}} \cdot h_0|, 0.25, 2), \tag{14}$$

where $h_{\text{rot}}$ and $h_0 = (0.0, 1.0, 0.0, 0.0)$ are the current and desired rod orientation as quaternions, respectively. Lastly, we utilize the following error

$$\text{err}(q, \dot{q}) = \sum_{i \in \{i| \ |q_i| > |q_i^b|\}} (|q_i| - |q_i^b|) + \sum_{j \in \{j| \ |\dot{q}_j| > |\dot{q}_j^b|\}} (|\dot{q}_j| - |\dot{q}_j^b|). \tag{15}$$

Here, $q, \dot{q}, q^b$ and $\dot{q}^b$ are the robot joint's position and velocity as well as their respective bounds. Additionally, we apply an action cost at each time step $t$

$$\tau_t = 5 \cdot 10^{-4} \sum_i^K (a_t^i)^2, \tag{16}$$

where $K = 7$ is the number of DoF. In total we consider three different rewards:

**Dense Reward.** The dense reward provides information about the goal and rotation distance in each time step $t$ on top of the utility rewards

$$R_{\text{tot}} = -R_{\text{rod}} - R_{\text{rod\_rotation}} - 5e^{-4}\tau_t - \text{err}(q, \dot{q}) - 3.5 R_{\text{goal}} - 2 R_{\text{rotation}}. \tag{17}$$

**Temporal Sparse Reward.** The time-dependent sparse reward is similar to the dense reward, but only returns the goal and rotation distance in the last time step $T$

$$R_{\text{tot}} = \begin{cases} -R_{\text{rod}} - R_{\text{rod\_rotation}} - 0.02\tau_t - \text{err}(q, \dot{q}), & t < T, \\ -R_{\text{rod}} - R_{\text{rod\_rotation}} - 0.02\tau_t - \text{err}(q, \dot{q}) - 350R_{\text{goal}} - 200R_{\text{rotation}}, & t = T. \end{cases} \tag{18}$$

### B.3 HOPPER-JUMP

**Task Description.** In the Hopper jump task the agent has to learn to jump as high as possible and land on a certain goal position at the same time. We consider five basis functions per joint resulting in a 15-dimensional weight space. The context is four-dimensional consisting of the initial joint angles $\theta \in [-0.5, 0]$, $\gamma \in [-0.2, 0]$, $\phi \in [0, 0.785]$ and the goal landing position $x \in [0.3, 1.35]$. The full observation space extends the original observation space from the OpenAI gym Hopper by adding the x-value of the goal position and the x-y-z difference between the goal point and the reference point at the Hopper's foot. The action space is the same as for the original Hopper task.

**Non-Markovian Reward.** In each time-step $t$ we provide an action cost

$$\tau_t = 10^{-3} \sum_i^K (a_t^i)^2, \tag{19}$$

where $K = 3$ is the number of DoF. In the last time-step $T$ of the episode we provide a reward which contains information about the whole episode as

$$\begin{aligned} R_{\text{height}} &= 10h_{\text{max}}, \\ R_{\text{gdist}} &= ||p_{\text{foot},T} - p_{\text{goal}}||_2, \\ R_{\text{cdist}} &= ||p_{\text{foot,contact}} - p_{\text{goal}}||, \\ R_{\text{healthy}} &= \begin{cases} 2 & \text{if } z_T \in [0.5, \infty] \text{ and } \theta, \gamma, \phi \in [-\infty, \infty] \\ 0 & \text{else} \end{cases}, \end{aligned}$$

where $h_{\text{max}}$ is the maximum jump height in z-direction of the center of mass reached during the whole episode, $p_{\text{foot},t}$ is the x-y-z position of the foot's heel at time step $t$, $p_{\text{foot,contact}}$ is the foot's heel position when having a contact with the ground after the first jump, $p_{\text{goal}}$ is the goal landing position of the heel. $R_{\text{healthy}}$ is a slightly modified reward of the healthy reward defined in the original hopper task. The hopper is considered as 'healthy' if the z position of the center of mass is within the range $[0.5m, \infty]$. This encourages the hopper to stand at the end of the episode. Note that all states need to be within the range $[-100, 100]$ for $R_{\text{healthy}}$. Since this is defined in the hopper task from OpenAI already, we haven't mentioned it here. The total reward at the end of an episode is given as

$$R_{\text{tot}} = -\sum_{t=0}^T \tau_t + R_{\text{height}} + R_{\text{gdist}} + R_{\text{cdist}} + R_{\text{healthy}}. \tag{20}$$

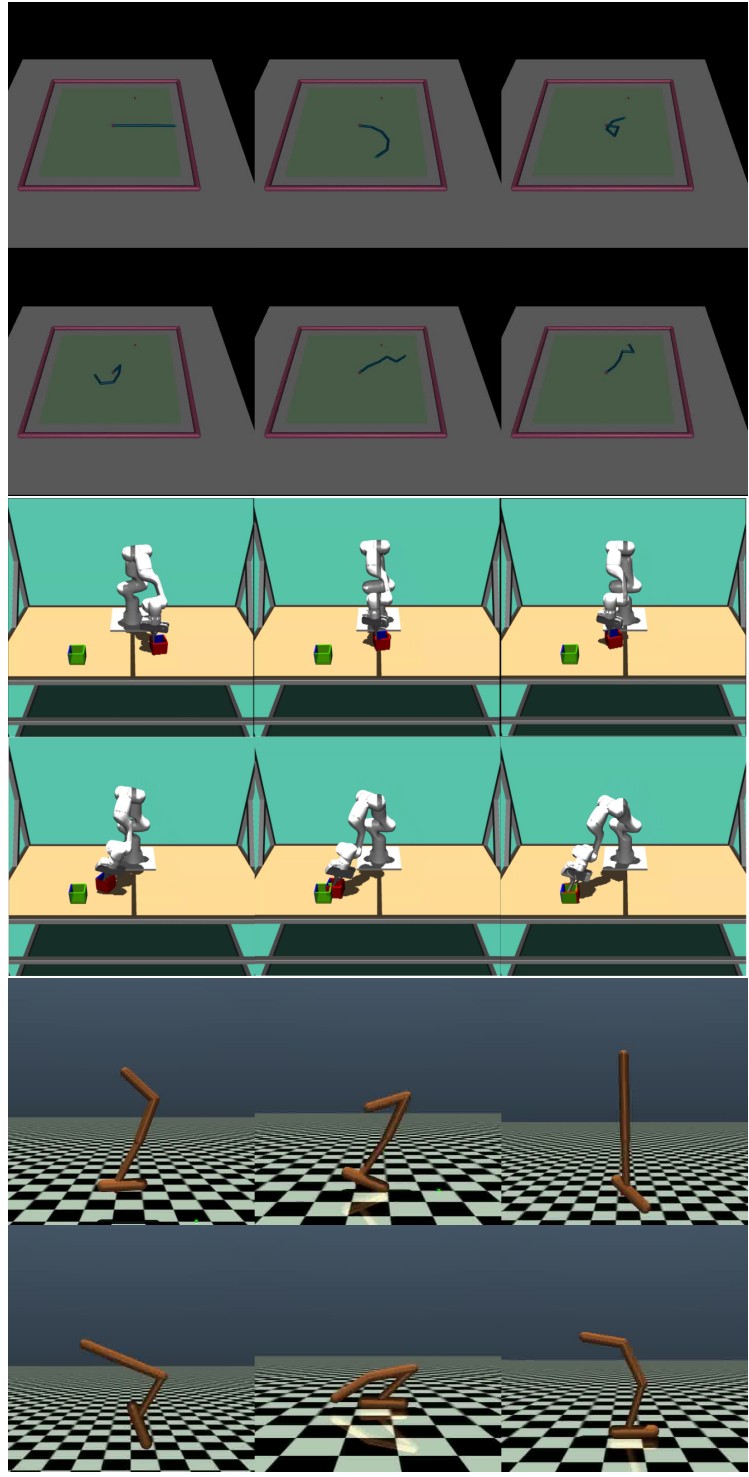

Figure 7: Example episodes for the 5D Reacher (top), Box Push (middle) and Hopper Jump tasks (bottom).

## C HYPERPARAMETERS

In this appendix we list the hyperparameters we used. The baseline results were taken from Otto et al. (2023b) and we thus refer to their paper for the corresponding hyperparameters. Unless noted otherwise, the same hyperparameters were used for all experiments.

Table 1: Hyperparameters for experiments with the Episode Transformer.

| Hyperparameter | 5D Reacher | Box Push | Hopper Jump |
|---|---|---|---|
| batch size | 64 | 64 | 64 |
| $n_{\text{seed}}$ | 500 | 500 | 500 |
| $n_{\text{pretrain}}$ | 1000 | 1000 | 1000 |
| $n_{\text{collect}}$ | 1 | 1 | 1 |
| $n_{\text{update}}$ | 5 | 5 | 5 |
| trust region loss weight $\beta$ | 10 | 25 | 25 |
| trust region mean bound $\epsilon_\mu$ | 0.05 | 0.005 | 0.005 |
| trust region covariance bound $\epsilon_\Sigma$ | 5e-4 | 5e-5 | 5e-5 |
| old policy decay rate $\tau$ | 0.995 | 0.995 | 0.995 |
| model learning rate | 1e-4 | 1e-4 | 1e-4 |
| model activation function | ReLU | ReLU | ReLU |
| model hidden size | 128 | 128 | 128 |
| number of Transformer blocks $N$ | 4 | 4 | 4 |
| number of multi-head attention heads $h$ | 4 | 4 | 4 |
| policy learning rate | 1e-4 | 1e-4 | 5e-5 |
| policy activation function | ReLU | ReLU | ReLU |
| policy hidden layers | [128, 128, 128] | [128, 128, 128] | [128, 128, 128] |
| policy initial std bonus | 1.0 | 1.0 | 1.0 |

