# OpenReview forum: "Episode Transformer: Model-based Episodic Reinforcement Learning"
_ICLR.cc/2024/Conference — ICLR 2024 Conference Withdrawn Submission_

### Official Review · Reviewer_Q5Ym · 2023-10-30

**Soundness:** 2 fair
**Presentation:** 2 fair
**Contribution:** 2 fair
**Rating:** 3
**Confidence:** 2

**Summary:**

This paper has proposed a model-based episodic reinforcement learning approach called Episode Transformer, and the proposed approach works in the contextual MDP setting. The Episode Transformer framework is built on the contexts and the movement primitives, and predicts the states and cumulative returns. Episode Transformer is able to improve the sample efficiency in several continuous control tasks with both sparse rewards and dense rewards.

**Strengths:**

The idea of using the Transformer structure to model state sequence is reasonable.

**Weaknesses:**

-	The writing of this paper is hard to follow. For example, the results in Figure 3 and Figure 4 can be placed in one large figure, as they demonstrates similar results in different tasks, so that the readers can compare the performance more easily.

-	As shown in Figure 6, the state prediction technique seems useless. However, this technique is core in model-based RL.

**Questions:**

1. Why does the variant method without state prediction work as well as the Episode Transformer method?

2. Figure 3 is strange, as I cannot see the NDP curve.

---

### Official Review · Reviewer_AjWh · 2023-10-31

**Soundness:** 2 fair
**Presentation:** 3 good
**Contribution:** 2 fair
**Rating:** 5
**Confidence:** 4

**Summary:**

This paper proposes using Transformer to learn a world model for Blackbox Reinforcement Learning where the policy is trained to generate parameterizations for complete trajectories rather than for each individual time step. The transformer's objective is to predict the cumulated episodic return and the environment states given the context and the desired trajectory of the agent. The authors suggest training the policy using the differentiable predicted reward and employing trust-region policy optimization to stabilize the training. The experimental results show improved sample efficiency in several robotic tasks.

**Strengths:**

- The idea is interesting and well-motivated
- The writing is generally easy to follow

**Weaknesses:**

- It is rather straightforward to use the idea of learning a world model with a Transformer. The novelty lies in the  Blackbox Reinforcement Learning setting, which introduced in prior works
-  The experiment result is not good. For example, in Fig. 3a, the proposed method performs worse than others. Overall, in term of reward performance, the final convergence of the method is not better than simply using trust region optimization. This weakens the main message of using Episode Transformer. There are some improvements in terms of sample efficiency. However, that comes without surprise due to using the world model
- The ablation study is not convincing. Fig. 6 (middle and right) shows no clear difference.

**Questions:**

- Using "episodic RL" may be confusing in another direction [1]. Perhaps using "Episode-based RL" like prior works is better
- Experiments: do you count the time of training world model when comparing sample efficiency?
- Can you show the learning curves of the world model?

[1] Gershman SJ, Daw ND. Reinforcement Learning and Episodic Memory in Humans and Animals: An Integrative Framework. Annu Rev Psychol. 2017 Jan 3;68:101-128. doi: 10.1146/annurev-psych-122414-033625. Epub 2016 Sep 2. PMID: 27618944; PMCID: PMC5953519.

---

### Official Review · Reviewer_in4T · 2023-10-31

**Soundness:** 2 fair
**Presentation:** 2 fair
**Contribution:** 2 fair
**Rating:** 3
**Confidence:** 3

**Summary:**

This paper proposed a Transformer-based model-based method (ET) for Episodic Reinforcement Learning (ERL). This method uses a Transformer encoder to capture the environment dynamic via predicting the per time-step states and the trajectory return. A separate policy module is learned with imagined returns from the Transformer world model to maximize the expected return. Thus, compared with model-free counterpart methods, this method is more data-efficient. During policy optimization, the gradient is allowed to back-propagate through the world model. A moving average of policy weights is maintained throughout training to stabilize the policy learning process, and the updated policy weight is also forced to keep close to the moving average (Trust Region Constraints, TRC). Ablation study necessitates the importance of TRC and stochastic back-propagation. However, predicting state per time-step seems not crucial. Experiments showed improved data efficiency compared with several baseline methods on the OpenAI gym tasks.

**Strengths:**

- The paper is well-written and easy to follow, though some details might need to be included.
- The data-efficiency statement is supported on the selected tasks.
- The ablation study is insightful and well-designed.

**Weaknesses:**

- How is the trajectory generator $\psi$ learned? Is the $\psi$ trained the same way for BB-TRPL and BB-PPO? How well is it trained?
- About the Trust Region Constraints (TRC):
	- I don't entirely agree with the statement, especially "In the step-based case, smaller errors during action selection can still be corrected at a later time step." In model-based RL, since the policy is learned during imagination, smaller errors are likely to be accumulated along rollouts instead of corrected at a later time step. This is known as the problem of compounding error.
	- I also doubt the argument behind TRC since the policy is trained with on-policy data during imagination, the learning policy is always in the "trust region". Moreover, equation 6 optimizes the expected return with respect to $\tilde{\pi}(w | c, \theta)$ instead of the learning policy. The policy is learned indirectly via the KL regularizer. I wonder if the learning is effective in this way.
	- The moving average of policy weights here reminds me of other actor-critic methods, so I'm curious about that, instead of using the TRC, maintaining a moving average of $R_\phi(c, w)$, directly optimizing the learning policy via $E_\pi(w | c)[- \tilde{R}_\phi(c, w)]$ might also help.
- Could the authors prove the necessity of using a Transformer by comparing the proposed method with an RNN-based world model? Especially when predicting the auxiliary state is not very important, I'm curious why not using a more lightweight RNN-based world model.

**Questions:**

- Is the $s_t$ a subset of $e_t$?
- What is the trajectory length of the target tasks? Are all the training trajectories having the same length? What is the transformer horizon set during the experiments? How does ET handle long-horizon tasks?

---

### Official Review · Reviewer_kg3S · 2023-11-01

**Soundness:** 2 fair
**Presentation:** 2 fair
**Contribution:** 2 fair
**Rating:** 3
**Confidence:** 3

**Summary:**

The paper discusses episodic RL, in which the actor predicts a parametrization of the agent's policy over the entire episode. The authors propose applying a transformer-based model (which they dub Episode Transformer) to this task by predicting episodic reward (and auxilliary state information) from desired states generated by motion primitives parametrized by the output of a learned policy.

This work builds upon prior work in episodic RL which applies the same scheme, using a trust region to learn a policy that outputs parameters for a motion primitive in the model-free setting.

**Strengths:**

Empirically the benefits of episodic RL for the tasks that are truly non-Markovian (e.g. high jump tasks) seem clear.

The method seems to give strong sample efficiency on the tasks studied.

The application of trust region layers seems to be somewhat novel in the MBRL setting and seems to be crucial for good performance.

**Weaknesses:**

The experiments section compares using only non-standard tasks, so it is somewhat difficult to tell whether the baselines are tuned appropriately. Specifically, the 5D reacher task seems quite simple (given that it is still a planar reaching problem) and I would expect most current RL methods to solve it to a similar level of performance under reasonable hyperparameter settings. The authors should add at least one "standard" environment to ensure fair comparison. It would be also good to compare against a modern sample-efficient RL method (e.g. REDQ, Dreamer, etc.) to validate the sample efficiency claims.

For the maximum jump height task, it may be helpful to include a baseline that keeps "max height achieved so far" as an additional state variable and applies SAC/PPO with the correct task reward (i.e. reward equal to $\max(0, h_t-\max_{1\le t \le t-1}h_t)$).

The ablations claim that ET works better with the stochastic backpropagation scheme than with REINFORCE, but the experiments don't seem to lend strong support for this and there seems to be quite a bit of overlap between the reported confidence intervals for the two. The authors should remove these claims or run a new experiment to tighten the confidence intervals and confirm whether there is some significant effect.

Given the ablations, it's not clear why it's necessary to predict true environment states in the first place. It seems that it would simplify the presentation of the method with no drawbacks if the state-prediction heads were removed. Additionally, as far as I can tell no ablation compares ET against a simpler model. This would be helpful to justify the Transformer model (e.g. can the same thing be accomplished with a smaller reward model?)

The introduction and related work should be reworked. Particularly, the explanation of Markovian rewards contains several questionable statements; e.g. the task of "throwing objects into bins" could absolutely be expressed with a Markovian reward - the problem in that case is with long-term credit assignment, in which case episodic RL could certainly perform better.

**Questions:**

Other comments:
 - The "smooth L1 loss" is the well-known Huber loss, which should be mentioned
 - TRPL = trust region projection layer, should be mentioned explicitly
 - The experiments all seem to cut off at different numbers of steps for each baseline
 - The formulation of the motion primitives for each task should be specified, at least in the appendix